# Cerebellar Astrocytes: Much More Than Passive Bystanders In Ataxia Pathophysiology

**DOI:** 10.3390/jcm9030757

**Published:** 2020-03-11

**Authors:** Valentina Cerrato

**Affiliations:** 1Department of Neuroscience Rita Levi-Montalcini, University of Turin, I-10126 Turin, Italy; valentina.cerrato@unito.it; Tel.: +39-011-670-6632; 2Neuroscience Institute Cavalieri Ottolenghi, Orbassano, I-10043 Turin, Italy

**Keywords:** cerebellum, ataxia, astrocytes, neurodegeneration, glia

## Abstract

Ataxia is a neurodegenerative syndrome, which can emerge as a major element of a disease or represent a symptom of more complex multisystemic disorders. It comprises several forms with a highly variegated etiology, mainly united by motor, balance, and speech impairments and, at the tissue level, by cerebellar atrophy and Purkinje cells degeneration. For this reason, the contribution of astrocytes to this disease has been largely overlooked in the past. Nevertheless, in the last few decades, growing evidences are pointing to cerebellar astrocytes as crucial players not only in the progression but also in the onset of distinct forms of ataxia. Although the current knowledge on this topic is very fragmentary and ataxia type-specific, the present review will attempt to provide a comprehensive view of astrocytes’ involvement across the distinct forms of this pathology. Here, it will be highlighted how, through consecutive stage-specific mechanisms, astrocytes can lead to non-cell autonomous neurodegeneration and, consequently, to the behavioral impairments typical of this disease. In light of that, treating astrocytes to heal neurons will be discussed as a potential complementary therapeutic approach for ataxic patients, a crucial point provided the absence of conclusive treatments for this disease.

## 1. Introduction

The role of the cerebellum in motor function is well recognized and results from an intricate circuitry through which this brain area gets connected with the basal ganglia and the cerebral cortex, as well as with peripheral motor and sensory pathways [1]. Malfunctions of any part of this circuitry result in imbalance and incoordination of posture and limbs, a disorder commonly known as ataxia and, specifically, as cerebellar ataxia in case of a cerebellar origin [2,3]. Nevertheless, besides motor alterations, ataxia has a very heterogeneous clinical presentation, comprising ocular impairments such as eye fixation deficits and nystagmus, and cognitive impairments with variable degrees of severity. This is due to the functional anatomy of the cerebellum, according to which distinct lobules in both the vermis and hemispheres are connected with distinct brain areas and are involved in diverse motor and non-motor functions [4]. Based on this functional anatomy, cerebellar ataxia is now clinically subdivided into (i) cerebellar motor syndrome, (ii) vestibulocerebellar syndrome, and (iii) cerebellar cognitive affective syndrome [5].

The ataxia syndrome can either emerge as the major element of a disease course or represent a symptom of more complex multisystemic disorders, and it can have a highly variegated etiology: this makes the proper classification and correct diagnosis of ataxic patients very challenging. Namely, ataxic disorders can be classified, according to their etiology, in acquired, hereditary, or sporadic non-hereditary degenerative forms, all comprising a huge list of diseases with distinct origins [3,6]. Acquired ataxias are due to exogenous or endogenous non-genetic causes, such as stroke, toxicity, vitamin deficiency, immune inflammation, nutritional deficiencies, and chronic central nervous system (CNS) infections [6,7]. Hereditary forms comprise several genetic disorders that are characterized by slowly progressive incoordination of movement and speech and can be inherited (i) in an autosomal recessive manner, such as Friedreich’s ataxia (FA) and Ataxia Telangiectasia (AT); (ii) in a dominant manner, like Spinocerebellar Ataxia (SCA) and Episodic Ataxia (EA); or (iii) in an X-linked manner [8]. Finally, sporadic ataxia can either be caused by random mutations in single gene or comprise those forms for which a definite genetic impairment or acquired etiology cannot be found, such as the cerebellar variant of the Multiple System Atrophy (MSA) [9] and the so-called sporadic adult-onset ataxia of unknown etiology (SAOA) [10]. 

Common features of most ataxias are cerebellar atrophy and Purkinje cells (PCs) degeneration. For this reason, despite back in the past gliosis was repeatedly reported in autopsies of human ataxic samples [11,12,13], researchers have focused for many years mainly on neuronal dysfunctions, largely overlooking the possible implication of non-neuronal cells, including microglia, oligodendrocytes, and astrocytes, in this disorder. Nevertheless, in the past years, this trend changed its direction. Indeed, early microglial activation was reported during post-mortem analyses in several forms and animal models of ataxia and was described to play a detrimental role by driving neuroinflammation [14]. Moreover, white matter degeneration was reported in SCA2,3,7,10 and AT [15,16,17,18,19,20,21] and often correlated with ataxia severity: although white matter changes have largely been suspected to be secondary to neuronal loss, they are also likely to result from oligodendrocytes impairments. Indeed, mutant *ATXN3* expression was shown to cause cell-autonomous transcriptional changes in oligodendrocytes that may directly disrupt white matter and be, therefore, implicated in the pathogenesis of SCA3 [16]. Similar changes in oligodendrocyte genes transcription may also occur in other genetic forms of ataxia and, in general, may cause oligodendrocyte dysfunctions that could impede the correct propagation of neuronal action potentials and, in turn, promote neuronal dysfunction and degeneration. Along the same line, in MSA, alpha-synuclein expression in oligodendrocytes was suggested to perturb their ability to provide trophic support to neurons, thereby contributing to diffuse neurodegeneration in this disease [22]. Lastly, an increased number of oligodendrocytes without mitotic activity was observed in childhood ataxia with diffuse central hypomyelination (CACH) syndrome, again suggesting that defects in these cells may be implicated in the etiopathology of cerebellar dysfunctions [23].

Among glial cells, astrocytes play a plethora of roles essential for brain development, homeostasis, and function [24,25,26]. As a consequence, no wonder they are increasingly being described as implicated in several neurodevelopmental and neurodegenerative disorders, such as Rett syndrome, fragile X mental retardation, amyotrophic lateral sclerosis, and Alzheimer’s disease [27,28,29], and they may be similarly involved in the pathophysiology of ataxia.

In the cerebellum, astrocytes are generated through a tightly regulated process and can be classified according to their morphologies and layering into three main categories, comprising Bergmann glia (BG) and granular layer astrocytes in the cerebellar cortex and fibrous astrocytes in the cerebellar white matter [30,31]. In the cortical layers, different astrocyte types interact with distinct neuronal subsets (Figure 1) and, therefore, are likely to develop neuron-specific functional properties, similarly important for the correct cerebellar development and functioning. Nevertheless, provided their highly specialized morphology and tight connection with the whole dendritic tree and soma of PCs, BGs are the most extensively studied astrocytes in the cerebellum, and their functions in supporting cerebellar development, PC synaptogenesis, and synaptic activity have already been described [32]. On the other side, very little is known about the specific functions and physiology of astrocytes in the granular layer, although their close relationship with the cerebellar glomeruli (i.e., the synaptic structures composed of mossy fibers rosettes, Golgi neuron boutons, and granule cells “GC” dendrites) [33,34] suggest that they may be similarly crucial for the regulation of tissue homeostasis and cerebellar circuits functioning.

In the last few decades, growing evidences are pointed at cerebellar astrocytes, specifically BG, as leading characters of a non-cell-autonomous neuronal degeneration in distinct forms of ataxia. Although intense work is still needed to clarify the role of astrocytes other than BG in this pathology, and despite the available knowledge is still very fragmentary and mainly ataxia type-specific, the present review will attempt to provide a comprehensive view of the mechanisms through which these cells are involved in the pathophysiology of ataxias, depicting a potential temporal cascade of events that eventually lead to the non-cell-autonomous neurodegeneration typical of this disease. A specific attention will be given to the pitfalls of the available models to study astrocytes’ contribution to the pathology, to promising new approaches to address this issue in the future and, importantly, to the potential forthcoming benefits of new therapies that specifically target astrocytes’ dysfunctions, a crucial point provided the absence of conclusive treatments for this disease. 

## 2. Astrocytes Impairments are often Associated to Ataxic-Like Symptoms and Ataxia

A first hint of the possible involvement of astrocytes in ataxia is the evidence that their deficiency at multiple levels, from developmental impairments to the lack of their functional support to neurons, often results in motor impairments (Table 1). Indeed, astrocytes ablation from either the developing or the adult murine brain through distinct approaches was observed to result in severe ataxia, associated with PCs ectopic layering and abnormal dendritic arborization and with the degeneration of GCs [35,36]. These results provided the first in vivo evidence that astrocytes are critical for both cerebellar development and neuronal survival in the adult mouse brain. Along the same line, not only the lack of BG specification from cerebellar radial glia, due to the manipulation of several distinct genes, resulted in abnormal foliation and lamination and in locomotion defects [37,38,39,40] but also the incorrect maintenance of the BG phenotype (i.e., their ectopic location in the molecular layer and/or acquisition of a stellate morphology) during late postnatal development caused signs of ataxia in diverse mouse models [41,42]. Further, morphological defects of BG radial processes in *vimentin*-null mice also translated into impaired motor coordination, likely due to an aberrant structural and homeostatic support on PCs [43]. At last, the acquisition of an immature- or reactive-like state by cerebellar astrocytes in *Dicer*-deficient mice, as revealed by alterations in their transcriptome profiles, was shown to precede and to be likely responsible for a rapid neurologic decline, characterized by ataxia, GCs apoptosis and PCs degeneration [44]. Interestingly, this non-cell-autonomous neurodegeneration was faster in the central and posterior zones of the cerebellum and slower in the anterior and nodular areas, thereby suggesting a diverse susceptibility of neurons to astrocytes dysfunctions or an intrinsic heterogeneity of astrocytes in their response to ataxia-related mutant gene expression.

Evidence showing that cerebellar astrocytes may be involved in the pathophysiology of ataxia also came from the study of distinct animal models of this and other pathologies. Reactive astrocytosis and BG proliferation were indeed associated with PCs degeneration in the first mouse model of SCA1 (a polyglutamine spinocerebellar ataxia due to the expansion of a CAG trinucleotide repeat in the *ATXN1* gene), in which the human mutant gene was expressed under the control of a PC-specific promoter [45]. On the other side, BG appeared to decrease before PCs degeneration in a different, non-PC-specific, mouse model of the same disease [46]. Moreover, the first three weeks of postnatal development were reported to be a critical period for the mutant *ATXN1* gene expression to permanently influence the symptoms observed in the adult [31,47]. It is interesting to note that this developmental stage corresponds to the peak of astrocytes proliferation and differentiation, thereby suggesting that altered astrocytes development due to mutant *ATXN1* expression may significantly contribute to disease progression. Increased expression of the glial fibrillary acidic protein (GFAP), astrogliosis, and thickening of BG processes were also observed in regions with extensive PC loss in a PC-specific *Tmem30a* KO mouse model that displays early-onset ataxia. In this case, astrocytic reactivity resulted from neuronal impairments and/or degeneration and likely further exacerbated neuronal loss contributing to disease progression, as will be discussed later [48]. A prominent astroglial activation and an impaired functioning of the glutamate-aspartate transporter GLAST, mediated by the proinflammatory cytokine interleukin-1β (IL-1β), were observed in the cerebella of experimental autoimmune encephalomyelitis (EAE) mice. These data hint that cerebellar astrocytes may also be involved in the cerebellar deficits and motor impairments typical of Multiple Sclerosis [49].

Interestingly, cerebellar astrocytes are also likely to be involved in the pathogenesis of those acquired ataxias due to alcohol abuse. Indeed, they were shown in vitro to be capable of synthesizing high levels of retinoic acid (RA) when exposed to ethanol [50]. RA, ligand for a group of ligand-activated transcription factors known as the RA receptors, is necessary for the development of many organs including the CNS but, when too much concentrated, can have teratogenic effects. Specifically, RA exposure was shown to cause abnormalities similar to those observed in the fetal alcohol syndrome (FAS), many of which are associated with cerebellar deficits: the aforementioned evidence thereby suggests that the cerebellar pathology exerted by ethanol may occur, at least in part, through an increased production of RA by cerebellar astrocytes. Although a direct in vivo demonstration of this hypothesis is still missing, the same authors showed that cerebellar RA levels were increased also in rat pups after ethanol administration [50]. Besides the dramatic effects of ethanol exposure on cerebellar development, alcohol abuse in adult life also causes cerebellar dysfunctions [51]. In this frame, astrocytes are again likely to play a crucial role. Indeed, even a brief ethanol exposure was shown to be sufficient to cause long-lasting changes in the gene expression, proliferation, and activity of mature astrocytes of distinct brain regions, with dramatic effects on the surrounding neurons [52]. Overall, these results indicate that astrocytes may contribute to cerebellar dysfunctions resulting from alcohol exposure both during fetal/postnatal and adult life.

## 3. Astrocytes Involvement in Ataxia as Revealed by Astrocyte-Specific Mutant Animal Models

In light of the aforementioned evidence and of those implying astrocytes as involved in the pathogenesis of several neurologic disorders, over the last fifteen years, scientists have begun to shift gradually their focus from neurons to astrocytes in the investigation of the cellular mechanisms underlying ataxia. Due to the genetic nature of many forms of this disease, the creation of animal models in which the mutant genes were selectively expressed or deleted in glial cells or, specifically, in astrocytes has been the most common approach to tackle this issue.

A first example is represented by SCAs that mostly belong to the category of polyglutamine diseases and comprise distinct forms characterized by CAG expansions on specific genes. The glial expression of SCA-related polyglutamine proteins has been shown several times to cause non-cell autonomous neurological symptoms, pointing to glial cells as anything but passive bystanders in the pathogenesis of this disease. Drosophila models expressing in glial cells the mutant *ATXN1* and *ATXN3* genes, associated with SCA1 and SCA3, respectively, showed accumulation of the protein fragments in glial intranuclear inclusions, accompanied by progressive degeneration of glial and neuronal cells, severe behavioral defects, and lethality [53,54]. Similar pieces of evidence of non-cell autonomous neurodegeneration were also obtained in a SCA7 mouse model where the expression of the mutant *ATXN7* gene in all cells but not in PCs led to ataxia and PCs degeneration [55]. Nevertheless, the demonstration that astrocytes were the main actors in this process only came some years later, when the conditional expression in BG of the mutant gene through a *Gfa2* promoter (version of the *GFAP* promoter with restricted expression in BG) was sufficient to cause motor incoordination and PCs pathology [56]. Later, the same authors developed an elegant animal model to better investigate the contribution of BG to the disease pathogenesis, by removing the mutant gene, in a context of ubiquitous expression, in a temporally and cell type-regulated fashion. Although the deletion of the mutant *ATXN7* gene solely in BG was not sufficient per se to prevent the development of ataxia and neurodegeneration, its excision from BG in combination with PCs and inferior olivary (IO) neurons yielded a synergistic effect on the delay of symptoms onset [57]. Altogether, these results support a significant role of BG in SCA7 pathogenesis where both cell-autonomous and non-cell autonomous mechanisms are likely to contribute to neurodegeneration in a tightly interconnected network between neurons and BG. Similarly, neurodegeneration in SCA17 was reported to depend on the concomitant expression of the mutant *TATA-binding protein (TBP)* gene in both neurons and astrocytes, while the mutant gene expressed in either of the two cell populations only caused a mild PCs degeneration [58]. A synergistic effect of defective neurons and astrocytes holds true also for other forms of ataxia, as addressed later in this review.

Another genetic form of ataxia is Friedreich ataxia (FA), caused by an unstable GAA expansion in the first intron of the *FXN* gene, which results in decreased levels of a protein called frataxin. To mimic frataxin deficiency, animal models were developed in which the wild-type *FXN* gene was ubiquitously silenced, resulting in neuronal degeneration, motor impairments, and reduced lifespan. Interestingly, specific deletion of this gene in glial cells in Drosophila generated FA-like symptoms comparable to those of the whole-body knockout flies [59], and its ablation in developing mice from GFAP-expressing neuronal and astrocyte precursors resulted in severe ataxia and early death, associated with growth and survival impairments in cerebellar astrocytes [60]. Although, in this mouse model, the contribution of mutant neuronal and astrocyte precursors could not be discriminated, the evidence that the alterations observed were specific for cerebellar astrocytes and absent in forebrain astrocytes suggests that a greater vulnerability of developing cerebellar astrocytes to frataxin depletion may contribute to the highly specific cerebellar disturbances typical of FA. On the other side, in one of the very few studies exploiting cultured human astrocytes, cortical astrocytes with frataxin deficiency (obtained through an shRNA knockdown approach) also showed detrimental effects, with signs of reduced proliferation and survival. This hints that the cerebellar-specific phenotype of this disease may also result from minor sensitivity to frataxin-deficient astrocytes of some neuronal populations in regions other than the cerebellum [61].

Ataxia Telangiectasia is a rare inherited disorder caused by mutations in the *Ataxia Telangiectasia Mutated (ATM)* gene, coding for the homonym protein involved in cell division and DNA repair. The first evidence of a glial-mediated non-cell autonomous mechanism of neurodegeneration in AT came from the specific *ATM* knockdown in glial cells in Drosophila, shown to be sufficient to activate an inflammatory response mediated by glial cells that drove the degeneration of both neurons and glia and resulted in reduced mobility and decreased lifespan [62]. Nevertheless, the aberrant effects of *ATM* deficiency in mouse cerebellar astrocytes were unveiled with an in vitro chimeric system and confirmed in vivo only very recently [63]. Indeed, while chimeric cultures composed of normal astrocytes and *ATM*-deficient neurons did not show any structural or functional anomalies, *ATM*-deficient astrocytes co-cultured with normal neurons not only showed a reduced length and number of processes but also led to altered numbers of pre- and postsynaptic puncta, to impaired neuronal synchronization, and to significant neuronal death. Adult cerebella from *ATM*-depleted mice similarly showed a disrupted morphology of both BG and GL astrocytes, with increased expression of synaptic markers at GABAergic synapses [63]. Although the neuronal population in the chimeric primary cultures was mostly represented by GCs and therefore no evidences are available on the effect of *ATM*-deficient astrocytes on PCs, on the whole, these results suggest that the neurodegeneration and impaired network dynamics in AT may at least in part due to a malfunctioning of cerebellar astrocytes, resulting in aberrant structural connections among neurons.

Niemann–Pick disease type C (NPC) is a rare autosomal recessive multisystemic disorder with early-onset ataxia as a cardinal feature, due to a progressive loss of cerebellar PCs [64]. The gene underlying 95% of the cases of this disorder is *NPC1*, involved in late endosomal lipid sorting and trafficking and, therefore, exploited as the main target for mutations or deletion in animal models intended to mimic this pathology. The available evidences on the role of glial cells and/or astrocytes in the coordination problems and progressive cerebellar degeneration typical of NPC have been controversial but eventually pointed to astrocytes as crucial contributors to this disease. Exploiting a chimeric mouse model with functional *NPC1* only in some cells with a mosaic-like pattern, Ko and colleagues [65] first concluded that degeneration of PCs was cell-autonomous, similarly to Lopez et al. [66] who, some years later, observed that *NPC1* rescue solely in astrocytes was not capable of preventing neurodegeneration and motor incoordination in *NPC1*-null mice. Nevertheless, these evidences have some points of criticism and are not sufficient at all to rule out astrocytes’ contribution to NPC pathology. Indeed, provided the tight connection of individual PCs with up to eight BG, in a mosaic context the lack of rescue of mutant neurons by some surrounding wild type astrocytes and, vice versa, the absence of detrimental effects exerted by some mutant astrocytes on healthy PCs [65] may be due to compensatory effects by other mutant or normal astrocytes, respectively. To clarify this point, animal models in which the *NPC1* gene is conditionally mutated in all astrocytes are needed. On the other side, although *NPC1* reintroduction in neurons of *NPC1*-deficient mice prevented neuron degeneration, this resulted in an only temporarily improved motor coordination, eventually not sufficient to block disease progression and mouse death [66]. Although the authors did not comment on this, this temporary rescue is likely to be due to surrounding mutant astrocytes that eventually contribute to the progression of the ataxic phenotype. Along this line, if neither astrocyte- nor neuron-specific *NPC1* rescues are per se sufficient for an efficient recovery, it is likely that both neurons and astrocytes are involved in the pathogenesis of this disease. Indeed, an analogous study, in which the astrocyte-targeted re-expression of *NPC1* in *NPC1*-null mice was obtained through a *GFAP* promoter-driven transgene, reported an enhanced survival of mice, decreased astrocytes reactivity, and delayed degeneration of PCs [67], thereby indicating that astrocytes contribute to disease progression. The same authors successively confirmed the co-operation of dysfunctional astrocytes and neurons in NPC by re-expressing the normal gene in both cell types, that translated into a nearly normal phenotype with neurodegeneration, motor symptoms, and death all delayed of several months compared to the neuron-only re-expression [68].

Further positive evidences came also from in vitro studies. First, in mouse neuron–astrocyte cocultures, *NPC1*-null astrocytes were shown to cause a decreased neurite growth of wild-type neurons due to a reduced estradiol release [69]. Moreover, simultaneous absence of the gene in co-cultured neurons and glial cells, but not in neurons solely, resulted in impaired presynaptic development, as revealed by reduced synaptophysin-positive puncta on PC dendrites [70]. Although it is not clear whether this altered presynaptic input on PCs may contribute to the progressive degeneration of these cells and the behavioral impairments typical of NPC, these results indicate that both neuronal and glial *NPC1* are required for proper presynaptic development and, added to the previous observations, confirm that neurons and astrocytes may cooperate in NPC disease progression. To further confirm this point, the investigation of the in vivo effects of an astrocyte-and/or neuron-specific deletion of the gene is still missing and is needed in future studies.

Overall, these results confirm that the complex interplay between neurons and astrocytes is necessary for the proper development and functioning of the cerebellar circuitries; at least in genetic forms of ataxias, where astrocyte-specific contribution was so far investigated, impairments in neurons and astrocytes both contribute to disease progression. Some of the mechanisms underlying this process have already been clarified and will be hereafter addressed.

## 4. How do Astrocytes Contribute to Ataxias? A Multi-Step Hypothesis

### 4.1. Cell-Autonomous Effects of Mutant Gene Expression in Astrocytes

Mutant ataxia-related genes are ubiquitously expressed and, therefore, are unlikely to cause developmental and/or functional impairments in neurons solely. Rather, their expression in astrocytes is likely to act as a first hit with distinct pathogenetic ripple effects.

As a first example, in a human astrocyte cell culture model of SCA7, the mutant *ATXN7* gene was shown to downregulate the expression of *RELN*, gene intimately involved in the development and maintenance of the whole cerebellum and, specifically, of PCs [71]. Its downregulation in mutant SCA7 astrocytes was suggested to cause a non-cell autonomous, astrocytes-mediated, selective degeneration of PCS, thereby likely explaining the tissue specificity of SCA7. Nevertheless, these results still need to be confirmed in vivo in both the developing and mature cerebellum. Similarly, a SCA1 mouse model allowed to identify *MAXER*, coding for a novel multiple α-helix protein located at endoplasmic reticulum (ER), as an *ATXN1* target gene. MAXER reduction in BG was shown to mediate their functional deficiency, inhibiting their proliferation, and reducing GLAST expression. The reduced levels of GLAST, both due to a decreased total number of BG and to a reduced expression on the spared cells, in turn, translated in glutamate-mediated neurotoxicity on PCs that could be rescued by *MAXER* overexpression [46]. Reduced numbers of BG, due to their massive death, were also very recently described in a SCA28 mouse model, in which the mutated gene *AFG3L2* was selectively expressed in astrocytes [72]. Here, BG death was only the endpoint of a progressive degenerative process, characterized by their ectopic position, aberrant morphology, decreased expression of the calcium-binding protein S100 and the glutamate transporter GLT-1, upregulation of GFAP, abnormal mitochondrial network, and activated metabolic stress response. Altogether, these alterations may result in the lack of metabolic support on PCs, in glutamate-mediated PCs excitotoxicity and in the acquisition of a “reactive” phenotype by BG, and may, therefore, not only magnify but even cause PCs degeneration in SCA28. The acquisition of a reactive phenotype by astrocytes following their expression of mutant genes is indeed a common feature of distinct forms of ataxia. In the first humanized *ATXN3* knock-in mouse model (Ki91 mice), expression of the mutant *ATXN3* in astrocytes was reported to cause astrogliosis in the cerebellar white matter (one of the hallmarks of SCA3 in humans) due to the expression of *Serpina3n* [73]. Interestingly, *Serpina3n* expression was evident already in newborn mice, long before the onset of PCs degeneration and behavioral deficits in motor coordination, that appeared much later in one-year-old cerebella. Similarly, activated inflammatory pathways such as NF-kB were uncovered in mutant astrocytes in a cell culture model of SCA17 and their blocking was shown to ameliorate neurodegeneration [58].

Altogether, these evidences indicate that the expression of mutant ataxia-related genes in astrocytes may act as an intrinsic factor that, in turn, causes their developmental and/or functional impairments and ultimately contributes, or causes, degeneration of the nearby neurons.

One of the first and most common impairments appears to be the acquisition of a reactive phenotype by astrocytes whose detrimental effects in distinct forms of ataxia will be addressed in the next section.

### 4.2. Astrogliosis: An Early Point Of No Return?

Cerebellar astrogliosis is not only among the first cell-intrinsic effects of the expression of ataxia-related genes in cerebellar astrocytes [58,72,73] but it is also probably one of the astrocytes’ earliest responses to the abnormal behaviors of mutant neurons.

Early astrocyte activation was indeed observed in both the cerebellar nuclei (CN) and cortex in a SCA21 mouse model in which the mutated gene *TMEM240* was expressed only in neurons [74]. Here, neuronal lysosomal impairments likely led to an active release of Damage-associated molecular patterns (DAMPs) or other factors that in turn triggered astrogliosis, resulting in motor dysfunctions without signs of neurodegeneration. Similarly, cerebellar astroglia was reported in several SCA1 mouse models to exhibit early signs of reactivity, recognized through increased GFAP expression, hypertrophy of cell bodies and processes, and expression of pro-inflammatory mediators [75]. Interestingly, this occurred well before the first signs of neurodegeneration and motor impairments and correlated with the disease in a spatio-temporal pattern, with most of the activated cells found in the most severely affected cerebellar regions. Again, this astrocyte activation was not caused by neurodegeneration, rather, it was shown to be triggered by signals originating from mutant PCs and could be arrested by preventing neuronal expression of the mutant *ATXN1* gene. Successively, the same authors demonstrated that cerebellar astrogliosis contributes to SCA1 in a biphasic manner. By modulating astrocytes reactivity through the inhibition of the astroglial NF-κB (nuclear factor kappa-light-chain-enhancer of activated B cells) signaling, they showed that when this pathway was inhibited during an early stage of disease (i.e., before the onset of motor deficits), the disease severity was exacerbated, whereas its inhibition during a later phase ameliorated the motor impairments without, however, saving PCs from death [76]. These results are suggestive of the first neuroprotective role played by astrocytes in SCA1 pathogenesis, which becomes harmful later on. Very recently, an early increase in the expression of brain-derived neurotrophic factor (BDNF) was identified as a key mediator of the initial neuroprotective role of astrocytes: an early delivery of this factor is indeed capable of delaying motor impairments and PCs pathology in a SCA1 mouse model [77]. Whether a biphasic role of astrogliosis in the disease onset also holds true for other kinds of ataxia is still needs to be elucidated.

The crucial role of astrogliosis in the pathogenesis of ataxia has been recently further demonstrated in conditional mouse models of cerebellar neuroinflammation, where the *IκB kinase 2 (IKK2)*, activator of the NF-κB pathway, was expressed in astrocytes or selectively in BG [78]. *IKK2* expression for a limited time was indeed sufficient to cause neuroinflammation, astrogliosis, and subsequent ataxic features due to selective degeneration of PCs. On the other side, the inactivation of Myd88, a crucial factor for the activation of the IKK/NF-κB pathway, was shown to mitigate neuroinflammation and PCs death in a SCA6 mouse model [79].

Overall, these evidences demonstrated that astrocytes’ activation, induced either cell- or non-cell-autonomously, can exacerbate or even cause neuronal dysfunctions, which in turn trigger a further amplification of astrogliosis in a detrimental vicious circle. This astrogliosis-like astrocytes activation was shown to be irreversible in some instances [78], determining a very early and time-restricted “point of no return” that points to early intervention as a crucial approach for the development of new therapeutic strategies. Importantly, this pathogenetic mechanism is likely to be crucial not only for the genetic forms of ataxia so far described but also for the inflammatory cerebellar ataxias, occurring in various autoimmune/inflammatory conditions in which inflammatory insults cause selective cerebellar neurodegeneration.

### 4.3. Astrogliosis Detrimental Effects on the Surrounding Neurons

After brain injury, astrocytes were reported to undergo a dramatic morphological and transcriptional transformation and acquire two possible reactive phenotypes, named “A1” and “A2” based on their inductive cue and effects on neurons and other surrounding cells [80,81]. Specifically, while A2 astrocytes are induced by ischemia and strongly promote neuronal survival and tissue repair by secreting neurotrophic factors, A1 are induced after acute brain injury by microglia and, by secreting neurotoxins, are lethal to both neurons and oligodendrocytes. Moreover, A1 astrocytes fail in promoting neuronal survival, synapse formation, and function, and are unable to phagocyte synapses and myelin debris. These impaired mechanisms all result in neurodegeneration and often contribute to disease progression [82]. Although a direct evidence that reactive astrocytes in cerebellar ataxia acquire an A1 phenotype is still missing, several impairments were described to occur in these cells that may cause non-cell-autonomous neurodegenerative effects. These impairments, which will be addressed hereafter, comprise a deficient glutamate uptake, the loss of structural and functional support for neurons, and the production of toxic substances or their impaired clearance from the extracellular space.

#### 4.3.1. Neurotoxicity Caused by Impaired Glutamate Uptake

One of the main consequences of the morphological changes in reactive astrocytes is an altered allocation of GLAST, astrocytic glutamate transporter necessary to maintain extracellular glutamate concentrations and avoid neuronal excitotoxicity. GLAST mispositioning may either be due to its spatial redistribution following astrogliosis, like its cluster-like appearance reported to occur in a SCA1 transgenic mouse model [83] or from the activation-related retraction of astrocytes processes, both resulting in its dislocation from PCs synapses. Furthermore, astrogliosis induced by the activation in astrocytes of IKK2, enzymatic complex involved in propagating the cellular response to inflammation, resulted in a strong downregulation of both GLAST and GLT-1, indicating that IKK2 per se may be a negative regulator of their expression in vivo and may directly mediate the local imbalance of glutamate homeostasis and neuronal excitotoxicity, at least in inflammatory cerebellar ataxias [78]. A progressive loss of GLAST was also reported in a mouse model of SCA5, leading to PCs degeneration and the worsening of the ataxic phenotype [84]. Here, PCs death was the result of a synergistic loss first of the neuronal glutamate transporter Excitatory amino-acid transporter 4 (EAAT4) in PCs and then of GLAST in astrocytes, with PCs in the posterior cerebellum showing a higher susceptibility due to the differential patterns of expression of EAAT4 across parasagittal bands [85,86]. Although the molecular mechanisms leading to the progressive loss of GLAST in these mice still need to be elucidated, it is tempting to speculate that this may result from the induction of an astrocyte reactive phenotype by surrounding sick PCs.

The significant contribution of glutamate-associated pathways to the pathogenesis of ataxia was very recently confirmed through innovative disease models based on SCA2 or SCA3 patients-derived induced pluripotent stem cells (iPSCs) [87]. Indeed, the treatment of iPSCs-derived neurons with glutamate resulted in altered composition of glutamate receptors, intracellular Ca^2+^ imbalance and, eventually, cytotoxicity and degeneration.

It is important to note that, besides resulting from astrocytes reactivity, impairments in glutamate uptake may also be caused by cell-autonomous mechanisms. Indeed, the ablation in mice of the *SLC1A3* gene, coding for the GLAST transporter, results in severe motor impairments and several missense mutations, either inherited or *de novo*, in the *GLAST* gene were shown to cause distinct forms of episodic ataxia (EA), whose symptoms severity appeared sometimes correlated with the extent of glutamate transporter dysfunction (Table 2) [88,89,90,91,92,93]. Moreover, reduced expressions of GLAST and/or GLT-1 were reported in models of SCA1, SCA7, and SCA28 in which the mutant genes were selectively expressed in astrocytes [46,56,72], indicating that the genetic alterations underlying some forms of ataxia may directly interfere with the astrocytes-mediated glutamate uptake.

Interestingly, a defective expression of GLT-1 in BG, associated with PCs hyperexcitation and motor impairments, was also reported in a mouse model of myotonic dystrophy type 1 (DM1), a multisystemic disorder in which cerebellar abnormalities are implicated [94]. This evidence suggests that an imbalance in glutamate uptake by cerebellar astrocytes may also contribute to a broad spectrum of disorders in which the cerebellum is involved through the altered motor and/or non-motor functions.

**Table 2 jcm-09-00757-t002:** Mutations in the *SLC1A3 (Solute Carrier Family 1 Member 3)* gene cause episodic ataxia 6 (EA6). Distinct missense mutations, either inherited or sporadic, were identified in the *SLC1A3* gene in patients with EA. This syndrome was designated as episodic ataxia 6 (EA6), although it shared many overlapping clinical features with EA2, identified by the presence of heterozygous pathogenic variants in the *CACNA1A* (calcium voltage-gated channel subunit alpha1) gene.

Mutation in The *SLC1A3* Gene	Functional Implications	Clinical Features	Family History	Reference
Heterozygous de novo Pro290Arg missense mutation	Decreased expression of GLAST and reduced capacity of glutamate uptake; increased anion currents through GLAST	Episodic and progressive ataxia, seizures, alternating hemiplegia, and migraine headache	Sporadic	[89,95]
Cys186Ser missense mutation	Modest but significant reduction of glutamate uptake	Milder manifestations of EA without seizures or alternating hemiplegia; overall similar to EA2	+	[90]
Heterozygous Val393Ile missense mutation	May influence glutamate binding and anion conductance	EA2 like symptoms; recurrent ataxia, slurred speech, interictal nystagmus with late-onset age of sixth decade	+	[91]
Arg454Gln (Arg499Gln) missense mutation	n.a.	Progressive ataxia, dysarthria, dysphagia, in some cases with adult-onset	+	[96]
Missense Ala329Thr mutation (concomitant with a mutation in the *CACNA1A* gene)	n.a.	Ataxia, dizziness, gaze-evoked nystagmus, seizures	+	[92]
De novo Thr318Ala missense mutation	n.a.	Typical EA2-like symptoms: recurrent ataxia, slurred speech; interictal nystagmus, mild cognitive impairment. Late-onset in the fourth or sixth decades	Sporadic	[92]
De novo Met128Arg missense mutation	May perturb the hydrophobic status of the membrane and affect glutamate uptake	Repeated episodes of ataxia: truncal ataxia, intentional tremor, slurred speech	Sporadic	[93]

+, presence of family history; EA, episodic ataxia, GLAST, glutamate/aspartate transporter; n.a., not available.

#### 4.3.2. Neurotoxicity Caused by S100beta Release

Another mechanism through which reactive astrocytes contribute to PCs degeneration is the production and release of toxic substances, and the calcium-binding protein S100B turned out to be one of them. Indeed, the presence in PCs of S100B-containing vacuoles, well before the onset of motor impairments, was reported in a SCA1 mouse model [97] and was later on traced back to autophagic mechanisms and associated with alterations in the morphology of PCs dendritic spines [98]. The same authors further demonstrated, both in vitro and in vivo, that the presence of S100B-loaded vacuoles in PCs correlated with the absence of ATXN1 nuclear inclusions, thereby suggesting that glial S100B may modulate the solubility of the mutant ATXN1 protein and, in turn, mediate its toxicity in PCs [99]. Although the underlying signaling pathways are still not fully known, the authors speculated that the sustained supply of S100B by reactive astrocytes may activate in PCs some factors, such as the NDR2 kinase, which destabilize BG-PC contact sites and neurite growth and result in PCs degeneration. Moreover, S100B Ca^2+^-chelating properties may modulate sodium channels in neurons and cause their Ca^2+^ overload, contributing to both aberrant synaptic transmission and neurotoxicity [100,101]. It is also worth noting that an excessive extracellular content of S100B was reported to cause alterations in BG cortical organization and morphology, thereby further contributing to the weakening of PC-BG contacts and the redistribution of glutamate transporters, in another, detrimental, vicious circle [100].

Importantly, elevated S100B concentrations were found in the serum of SCA3 patients, although uncorrelated to the disease duration or severity [102]. These results suggest that S100B overload may be a common pathogenetic mechanism of distinct forms of ataxia and, importantly, point to the application of this protein as a peripheral marker of disease in ataxia.

Overall, these evidences indicate that much more than a neuronal issue lags behind neurodegeneration in ataxias. Conversely, astrocytes actively participate in the neurodegenerative process, causing and/or exacerbating neuronal dysfunctions. Although most of the pathogenetic mechanisms described above can derive from cell-autonomous astrocytes dysfunctions and individually contribute to disease onset or progression, here a temporal cascade of events was proposed in which astrocytes are active and crucial players in distinct phases of the ataxia course (Figure 2). If true, this multi-step hypothesis of astrocytes involvement in ataxia should be taken into account in view of developing new astrocytes-targeted and, ideally, stage-specific therapeutic approaches, as will be discussed later.

## 5. Other Pathogenetic Mechanisms of Astrocytes Involvement in Ataxia

### 5.1. Well beyond Glutamate Uptake: The Role Of Glast As Ion Channel And Its Implication In The Pathogenesis Of Ataxia

Besides the mechanisms so far described, which may be involved at distinct levels in a putatively multi-step progression of the disease, another way in which astrocytes have been recently associated to the pathogenesis of ataxia is through an abnormal anion conductance of GLAST that, although mostly known for glutamate transport, also plays an ancillary function as a chloride channel [103] (Figure 2). Specifically, the episodic ataxia 6 (EA6)-related point mutation P290R in the *SLC1A3* gene (encoding GLAST) was shown not only to reduce the number of GLAST transporters on the surface of mammalian cells and impair their glutamate uptake but also to cause the appearance of larger anion currents with or without the administration of external glutamate [95]. Moreover, this mutation caused episodic paralysis of Drosophila larvae when expressed in glial cells [104]. A very similar behavior, with paralysis and astroglial aberrant morphologies, was induced after the expression in Drosophila of a chloride-extruding K^+^-Cl^−^ cotransporter and was rescued by the expression of a Na^+^-K^+^-Cl^−^ cotransporter, which normally allows chloride to enter into cells [104]. These results suggested for the first time that, although a reduced glutamate uptake was thought the be the main pathophysiological process underlying episodic ataxia, the P290R mutation in the *SLC1A3* gene also causes abnormal chloride flow from astrocytes, contributing in this way to disease progression. These evidences were indeed very recently confirmed by the development of a transgenic mouse line, in which the mutation was carried by GLAST-expressing astrocytes solely (Slc1a3^P290R/+^ mice). This animal model was shown to suffer from epilepsy, ataxia, and showed cerebellar atrophy, thereby closely resembling the features of EA6 [105]. Specifically, these degenerative effects were unveiled to be triggered by an increased chloride efflux from BG, resulting in BG apoptosis and, subsequently, in a secondary loss of glutamate clearance and cerebellar atrophy.

The maintenance of a certain chloride concentration inside BG and, likely, inside astrocytes in general appears, therefore, to be a crucial point that had been so far underestimated. Not only chloride efflux from astrocytes may reduce their volume and dislocate their processes from synapses but it may also cause their death, therefore impairing their supportive role in neuronal functions and survival. Moreover, a reduced astrocytic chloride concentration may increase the driving force for GABA uptake of those transporters on astrocytes that cotransport both GABA and chloride, therefore attenuating GABA-mediated synaptic transmission.

Astrocytes were shown in the past to actively accumulate chloride upon activation. It will be fascinating to clarify whether any kind of alteration in chloride homeostasis may result also from the acquisition of a reactive phenotype by astrocytes in ataxia, and be therefore part of the multi-step progression of the disease proposed in the previous section. Similarly, whether an impaired GLAST chloride conductance is also involved in ataxias other than EA6 still needs to be elucidated.

### 5.2. When Astrocytes “Burn Out”: Stressed Astrocytes Cause Accumulation Of Oxidative Stress And Neurodegeneration

The role of astrocytes as antioxidant supporters for neurons is well accepted [106,107,108], and the loss of this support was often shown to lead to neuronal death in diverse neurodegenerative diseases [109,110]. Oxidative stress, associated with the activation of both mitochondrial and endoplasmic reticulum (ER) stress responses in astrocytes, was indeed shown to be involved in the pathophysiology of distinct ataxias (Figure 2).

First, in distinct in vitro models of Ataxia Telangiectasia, *ATM*-deficient astrocytes were shown to activate their oxidative and ER stress responses, which in turn translated in a senescence-like growth arrest through multiple Extracellular signal-regulated kinases 1/2 (ERK1/2)-dependent mechanisms, as demonstrated by the fact that treatments with the antioxidant N-acetyl-L-cysteine (NAC) not only restored their defective proliferation but also suppressed ERK phosphorylation/activation [111,112,113]. Stressed and senescent astrocytes, in turn, failed in providing their antioxidant support on neurons that eventually degenerated. Recently, the mechanism underlying this suboptimal antioxidant support of astrocytes was found in their impaired ability to import l-cystine and produce reduced glutathione (GSH), which is normally secreted by healthy astroglia in response to oxidative stress to provide additional support to neurons [114]. Specifically, stressed cerebellar astrocytes isolated from *ATM*-mutant mice showed decreased expression of the cystine/glutamate exchanger subunit xCT and of GSH reductase, which translated into reduced levels of both intracellular and secreted GSH. Consequently, when co-cultured with healthy neurons, these astrocytes provided insufficient GSH levels to support neuronal survival [114]. Nevertheless, the absence of neurodegeneration in AT experimental models did not allow so far to clarify whether this mechanism also holds true for astrocytes in vivo.

A similar involvement of oxidative stress in neurodegeneration was also proposed for Friedreich ataxia, in light of the abnormal levels of antioxidant enzymes found in the cerebella of frataxin-depleted mice [60]. In a Drosophila model of this disease, the expression of the mutant frataxin protein in glial cells was indeed shown to increase lipid peroxidation, inducing mitochondrial dysfunction and, in turn, accumulation of fatty acids and cell degeneration, which translated in a shortened lifespan and impaired locomotor performances [59]. Interestingly, co-expression in the glia of *frataxin*-deficient flies of *Glaz*, one of the Drosophila homologs of apolipoprotein D (ApoD), was sufficient to increase the lifespan and improve locomotor activity, likely due to its modulation of lipid composition and oxidation. The same authors later demonstrated that also an enhanced iron transport inside mitochondria was involved in the degenerative phenotype induced by frataxin depletion, as demonstrated by the fact that a reduction of mitoferrin, a mitochondrial iron transporter, significantly improved the motor impairments, while its overexpression enhanced them [115]. Further, Loria and colleagues [61] reported that *frataxin*-depleted cultured human astrocytes had altered mitochondrial morphology and suffered great oxidative stress, therefore supporting the idea that an increased mitochondrial iron content favors astrocytes oxidative stress.

Besides mitochondrial dysfunction, endoplasmic reticulum (ER) stress was recently reported as a novel and crucial player in the progression and etiology of FA. Indeed, in a Drosophila model of this disease, the downregulation in glial cells of Marf, a protein known to play crucial roles in mitochondrial fusion, mitochondrial degradation, and in the interface between ER and mitochondria, improved motor functions, neuronal degeneration, and lipid homeostasis through the suppression of ER stress [116]. Very similar results were obtained through ER stress reduction by means of two different chemical compounds, allowing the authors to conclude that the advantages conferred by Marf downregulation in glia were mainly due to its role in the mitochondrial-ER tethering.

Overall, these evidences support the role of astrocytes in both AT and FA, which is mediated by a progressive accumulation of either mitochondrial or ER stress, or both. Stressed astrocytes, in turn, fail to support and protect from oxidative stress their surrounding neurons, which eventually degenerate. Besides that, the analysis of the secretome from cultured human frataxin-deficient astrocytes also supported the hypothesis that stressed astrocytes may directly mediate neuroinflammation and, in turn, neurodegeneration through the release of specific cytokines involved in these processes [61].

## 6. Treating Astrocytes to Heal Neurons: New Promising Approaches For Ataxia Therapy

Provided the complex and extremely heterogeneous scenario of ataxias, no curative and confirmed pharmacological treatments are still available, neither for those forms with unknown etiology nor for those whose causes have already been clarified. The available treatments are supposed to ease symptoms and overall improve the quality of life of ataxia patients, comprising (i) physical and speech therapies to mitigate tremors, stiffness, and speech problems, (ii) globulines injections to boost the immune system in case of immune-mediated ataxias, and (iii) counseling or medications to alleviate the associated depression, anxiety, and sleep disorders. The only exceptions are played by those ataxias caused by nutritional deficiencies, such as a lack of vitamin E or coenzyme Q10, which respond well to supplements, and by some subtypes of episodic ataxia, which can often be successfully treated with the drug Acetazolamide, a carbonic anhydrase inhibitor supposed to prevent ataxic attacks by normalizing regional pH in the cerebellum and, in turn, by inducing physiologic changes in several types of ion channels [117]. In view of developing new therapies aimed at eradicating the disease from its roots, it is reasonable to think that targeting astrocytes’ dysfunctions, besides neuronal impairments, maybe a very promising approach, provided their critical involvement in the onset and/or progression of this disease. These putative astrocyte-targeted treatments should strictly depend on the nature of the disease and, ideally, should be stage-specific, provided the variegated mechanisms through which astrocytes have been reported to be involved in distinct forms and phases of ataxia.

The beneficial effects of treatments limiting astrocytes impairments were reported in distinct ataxia animal models and, in some instances, also in ataxic patients. As an example, treatments with *estradiol* were shown to delay the onset of neurological symptoms and increase PCs survival in Niemann Pick disease, counteracting its decreased secretion by *NPC1*-deficient astrocytes [69]. Similarly, in SCA7 patients, treatment with the histone deacetylase inhibitor trichostatin *A* could, at least in part, restore the defective reelin transcription in mutant astrocytes, promoting the accumulation of the mutant ATXN7 protein into nuclear inclusions and, therefore, slowing disease progression by reducing PCs neurotoxicity [71].

Inflammation and astrogliosis also appear as crucial players in disease progression. In this context, anti-inflammatory astroglia-based therapeutic approaches have been proposed for both SCA1 and SCA17 [58,76]. Yet, the evidence that intracerebroventricular injections of IL-1 receptor antagonist (IL-1ra) in EAE mice reduced motor deficits are very promising in view of responsiveness to anti-inflammatory treatments also in other forms of ataxia [49]. Importantly, although a bi-phasic role of astrogliosis in ataxia pathophysiology has been so far suggested only for SCA1, these treatments may need to consider the stage of disease progression to achieve therapeutic efficacy. In a first phase of the disease, when astrocytes’ reactivity may be protective [76], the inhibition of astroglial NF-κB signaling, likely effective at later stages [58], may indeed result detrimental, while BDNF delivery may delay disease onset fostering astrocytes’ defensive role against neurotoxicity, at least in SCA1 [77]. Similarly, even stimulating BG/astrocytes proliferation at this stage could be a viable therapeutic intervention [118].

The release of S100B by reactive astrocytes is one of the mechanisms through which these cells mediate the non-cell autonomous neurodegeneration of the surrounding PCs. In this context, the construction of substances that attenuate the activity of myo-inositol monophosphatase 1 (IMPA1), target of S100B whose activation is involved in the spines and dendrites abnormalities in PCs, has been suggested as potential therapeutic approach for SCA1 [98]. Similarly, treatment with the S100B inhibitory peptide TRTK12 was shown to improve motor performances in a model of SCA1 [99]. Importantly, both the activity of myo-inositol and the presence of S100B were proposed as glial-based biomarkers that could serve as indicators of disease progression and severity [75].

Besides S100B secretion, the impairment in glutamate uptake is the other main mechanism through which astrocytes contribute to neuronal toxicity and death. In this frame, in a mouse model of SCA28, increased expression of the glutamate receptor EAAT2 in astrocytes, promoting glutamate clearance, was obtained with the administration of the antibiotic ceftriaxone and ameliorated the ataxic phenotype [119]. Similarly, the cerebellar phenotype of myotonic dystrophy mice, due to spontaneous hyperactivity of PCs resulting in motor incoordination, was also ameliorated by ceftriaxone treatment [94], thereby suggesting that this kind of approach may be beneficial in distinct ataxic patients. Along the same line, anti-glutamate drugs were shown to reduce the death of SCA2- and SCA3-iPSCs-derived neurons [87], further indicating that glutamate-associated pathways are very promising drug targets across the variegated spectrum of ataxias.

Antioxidant therapies likely represent another valid approach to effectively ameliorate at least AT and FA, by reducing oxidative stress and supporting astrocytes’ functions in neuronal survival. Indeed, in AT in vitro models, NAC treatments were able not only to suppress the activation of stress and senescence pathways in mutant astrocytes [112] but also to restore normal GSH levels by circumventing the defects in l-cystine import in these cells, which were, therefore, again capable of supporting neuronal survival and neurite outgrowth [114]. Another approach to normalize GSH levels and overcome neuronal death in AT may be the drug-induced activation of the cystine/glutamate exchanger subunit xCT, already suggested for Parkinson’s disease and amyotrophic lateral sclerosis [120,121]. This kind of therapy would be, however, highly problematic, since it could result in overaccumulation of glutamate in the extracellular space causing neuronal toxicity. The direct delivery of NAC or other cysteine precursors, therefore, likely represents the safest solution among the antioxidant approaches. Along the same line, apolipoprotein D may provide an important therapeutic target, due to its modulation of lipid composition and lipid oxidation, as hinted by evidences obtained in a Drosophila model of FA [59]. On the other side, counteracting iron transport inside mitochondria may be another promising option to slow down neuronal degeneration in FA, and this could be achieved through mitoferrin depletion or iron chelation therapies [115,122]. Last, insulin-like growth factor I (IGF-I) injections were shown to protect neurons from frataxin deficiency in a non-cell autonomous fashion, both enhancing frataxin levels in astrocytes and potentiating their neuroprotective properties through the stimulation of the Akt/mTOR pathway [123].

Of particular relevance, gene therapy in the shape of antisense oligonucleotides (ASO) has been recently shown to attenuate the disease in a SCA3 mouse model by targeting not only neurons but also astrocytes [124]. Along the same line, a novel kind of synthetic antisense long non-coding RNAs capable of upregulating the expression of specific genes of interest, called SINEUPS, has been very recently shown to rescue defective frataxin expression and activity in a human cellular model of Friedreich’s Ataxia [125]: it will be interesting to evaluate in the future whether this approach can target both neuronal and non-neuronal cells. Gene therapy undoubtedly represents the new frontier for genetic diseases, and the precise targeting of cells of interest without side effects is one of the main goals of this innovative therapeutic approach: in view of its application in the genetic forms of ataxia, these cellular targets should be not only neurons but also astrocytes.

Importantly, in case cerebellar astrocytes are healthy because not primarily affected in ataxia etiology, they are likely to take a crucial part in the so-called “cerebellar reserve” [126]. The cerebellar reserve is indeed defined as the capacity of the cerebellum to compensate and restore function in response to pathology, and cerebellar astrocytes, with their endogenous neuroprotective effects, are likely to counteract neuronal damage and loss as described to occur in several other kinds of brain injury. This aspect is particularly relevant in view of planning new treatment strategies that could boost the astrocytes’ neuroprotective activity, at least in the initial phase of the disease when they maintain their structural and functional integrity. Astrocytes were indeed described to be able to (i) remove the excessive glutamate released by damaged or dying neurons and, therefore, protect other cells from excitotoxicity and neurodegeneration; (ii) take damaged mitochondria up from sick neurons defending them against mitochondrial dysfunctions; (iii) help attracting robust inflammation to sites of tissue damage in order to neutralize pathogens, clear debris, and promote tissue repair; (iv) express high levels of GSH in response to oxidative stress, which often results from an overproduction of reactive oxygen species (ROS) following injury [127]. These same mechanisms are likely to occur also in ataxia. Indeed, astrocytes astrogliosis was proposed to be neuroprotective in the first phases of SCA1 [76] and distinct treatments boosting specific astrocytes’ neuroprotective mechanisms were described to ameliorate the cerebellar phenotype of distinct forms of ataxia. Among them, the aforementioned drugs targeting glutamate-associated pathways may be beneficial in ataxic patients not only by counteracting astrocytes’ impairment in glutamate uptake but also by supporting healthy astrocytes in their neuroprotective activity. Similarly, the antioxidant therapies described before may facilitate astrocytes’ response to oxidative stress. Besides their neuroprotective effects, cerebellar astrocytes are likely to contribute to the “cerebellar reserve” also by sustaining compensatory responses in neural circuits, restoring lost connectivity in dysfunctional cerebellar networks. This was indeed described to occur in chimeric culture models of AT in which both structural and functional anomalies of ATM-deficient neurons were completely rescued by the presence of healthy astrocytes [63]. Moreover, in the wake of the recent evidence indicating that differentiated astrocytes can be triggered to change their fate and reprogram into neurons in the injured mouse cerebral cortex [128], if properly boosted, cerebellar astrocytes may represent a very promising source of new neurons in ataxic contexts, thereby incrementing the cerebellar reserve. This may be true specifically for BG. Indeed, evidence suggests that they belong to the same lineage of PCs, with whom they share the same origin in the embryonic ventricular zone and partially overlapping developmental trajectories [31]. Moreover, BG postnatal progenitors share striking similarities, both under the molecular and morphological point of view, with the embryonic outer radial glia (oRG) responsible for the generation of the majority of cortical neurons, are capable in vitro of producing neurospheres, and were shown to be capable of reprogramming into granule cells (GCs) following a postnatal depletion of GC precursors [129]. For these reasons, BG may represent a novel and abundant stem cell population in the adult cerebellum and appear to be the best cerebellar candidates to be triggered toward the PC fate, particularly promising in view treating cerebellar disorders.

## 7. Conclusions and Future Perspectives

The evidences summarized here indicate that astrocytes play pivotal roles in the pathogenesis of ataxias. Specifically, although the available knowledge is still fragmentary and mostly disease-specific, it looks evident that numerous mechanisms of astrocytes involvement in neurodegeneration are shared by diverse forms of ataxias and likely contribute to disease progression in a phase-specific fashion, of particular relevance in view of developing new astrocytes-targeted therapeutic approaches. Nevertheless, most of this knowledge relies on experimental models that mostly fail to faithfully recapitulate the human disease. For instance, the available animal models for AT do not show robust neurodegeneration or cerebellar atrophy, hallmarks of the human disease [130], while a mouse model of SCA1 in which the mutant *ATXN1* was expressed at the endogenous level only showed mild behavioral impairments [46]. Similarly, an insufficient reduction of frataxin in FA mouse models translated into the absence of anomalies in motor coordination [60,131,132]. These effects, partially due to the short lifespan of mice that does not allow the neurons to accumulate significant damage, were often overcome by creating new animal models with transgenes expression exceeding the endogenous levels, which in turn led to dramatic and exaggerated outcomes that again did not reflect reality [56,133]. Importantly, in view of unveiling the astrocyte-specific effects of mutant protein expression/deletion, conditional experimental lines were developed in which, however, the transgene induction was often not sufficiently robust and, most importantly, depended on the kind of promoter and system (i.e., Cre-loxP, Tet-On/Off) used [60]. Lastly, many studies addressing glial involvement in ataxia [53,59,62,104,115,116] have been, still recently, based on the Drosophila model that, although very suitable to this aim due to its limited number of glial cells with well-known positions and molecular markers, may lead to an oversimplification of this issue. To overcome all these limitations, the use of patient-derived iPSCs differentiated into astrocytes or in the form of three-dimensional organoids, or of diseased human glial chimeric mice will represent, in the next future, a very promising approach to draw more solid conclusions on the cellular and molecular mechanisms underlying astrocytes involvement in ataxias and to screen drugs for treatments, as done for many other neurodegenerative disorders [134,135]. Moreover, the future identification of molecular markers specific for the distinct cerebellar astrocyte phenotypes, hopefully possible thanks to high throughput -omic approaches, will allow not only to differentiate distinct astrocyte subtypes from patient-derived cells but also to isolate distinct astrocytes populations from the available animal models and to develop new conditional ones in view of elucidating, if any, the role of cerebellar astrocytes other than BG in ataxia pathophysiology.

## Figures and Tables

**Figure 1 jcm-09-00757-f001:**
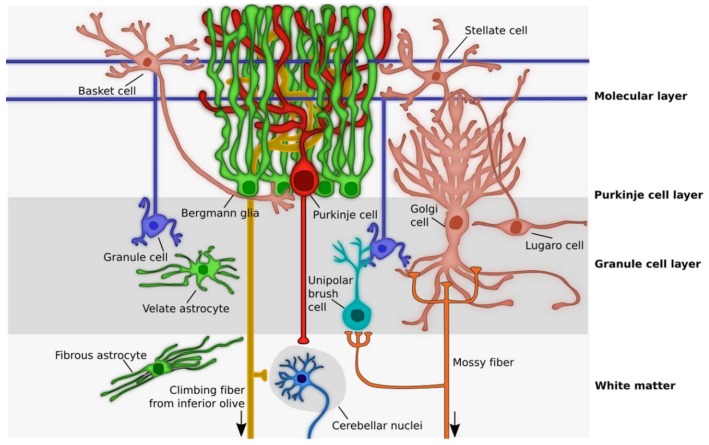
Schematic picture showing cerebellar cytoarchitecture. Cerebellar cortex is composed of three layers. In the innermost layer, the Granule cell layer, excitatory granule cells are surrounded by Golgi and Lugaro cells, two kinds of inhibitory interneurons, as well as by the excitatory unipolar brush cells and velate astrocytes. Here, velate astrocytes are in close relationship with the so-called cerebellar glomeruli, composed of mossy fibers rosettes, Golgi neuron boutons, and granule cells dendrites. The Purkinje cell layers host the cell bodies of Purkinje cells and Bergmann glia, whose dendrites and fibers, respectively, span the whole length of the outermost layer of the cerebellum, the Molecular layer. Here, granule cells’ parallel fibers synapse directly onto Purkinje cells dendrites and have contact points with molecular layer interneurons, called Basket and Stellate cells. Moreover, the processes of Bergmann glia are in tight connection with the whole dendritic tree and soma of Purkinje cells. In the cerebellar white matter, fibrous astrocytes are aligned to axons, while in the deep cerebellum the cerebellar nuclei neurons receive inputs from both Purkinje cells and climbing fibers of inferior olive neurons and project either back to the inferior olive (the GABAergic neurons) or to the brainstem, midbrain, and thalamus (the glutamatergic neurons). Arrow, projections coming from outside the cerebellum.

**Figure 2 jcm-09-00757-f002:**
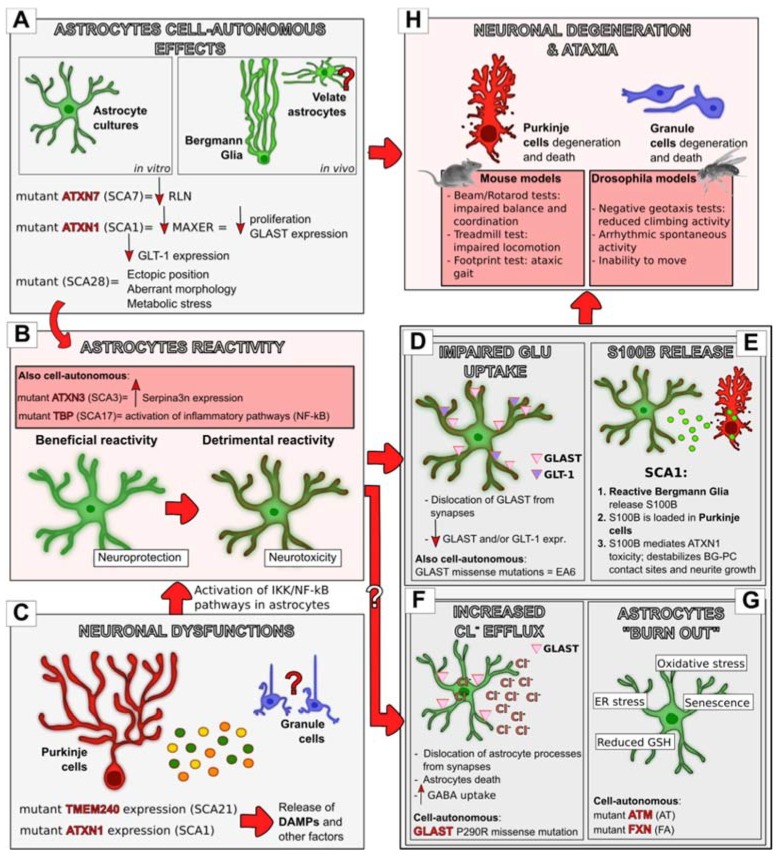
The multi-step hypothesis of astrocytes’ involvement in ataxia. Astrocytes contribute to neuronal degeneration and motor impairments in distinct phases of the disease course. The expression of ataxia-related mutant genes in astrocytes can cause cell-autonomous impairments (**A**) that can contribute to neurodegeneration either directly (**H**) or indirectly through the acquisition of a reactive phenotype (**B**). This was demonstrated to occur in distinct in vitro models, and, in vivo, in BG, while whether this happens also in the other cerebellar cortical astrocytes still needs to be clarified. Astrocytes’ reactivity can also result from neuronal impairments ((**C**); demonstrated for PCs, likely to occur also in GCs), through the release from damaged neurons of DAMPs and other factors that can activate IκB kinase (IKK)/NF-kB inflammatory pathways in astrocytes that, in turn, become reactive (**B**). Interestingly, astrocytes’ reactivity was shown to play a dual function in Spinocerebellar Ataxia (SCA1) progression, first beneficial and later detrimental. Whether this holds true also for other forms of ataxia still needs to be elucidated. Reactive astrocytes, in turn, trigger neurodegeneration through distinct mechanisms, like the impairment of glutamate uptake (**D**) and the release of the calcium-binding protein S100B (**E**). Other neurotoxic mechanisms were shown to be an increase in the chloride efflux from astrocytes (**F**) and their accumulation of stress responses (**G**). Both these mechanisms were described to result from cell-autonomous impairments in astrocytes but may also result indirectly from their acquisition of a reactive phenotype. Glu, Glutamate, Big red arrows, successive step; small arrows, increase (facing up) or decrease (facing down). White question mark, hypothetical successive step; red question mark, hypothetical cellular involvement.

**Table 1 jcm-09-00757-t001:** Astrocytes developmental or functional deficiencies result in neuronal degeneration and/or motor impairments.

Astrocytes Alterations	Animal Model Applied	Cellular and Histological Impairments	Behavioral impairments	Reference
Astrocytes ablated postnatally	Herpes simplex virus- thymidine kinase expressed in mice under the control of the *hGFAP* gene promoter	Disordered radial glia; PCs ectopically distributed; abnormal PCs dendritic trees, depletion of GCs; reduction in cerebellar size; disruption of the cellular layers	Severe ataxia	[35]
Astrocytes ablated in the adult	Targeted E. coli nitroreductase expression to the astrocytes of transgenic mice with the *GFAP* promoter	Abnormal PCs dendrites; GCs degeneration	Ataxic behavior	[36]
Absence of functional BG	Overexpression of the group C protein Sox4 in transgenic mice under the control of the hGFAP promoter	Fissures were not formed; neuronal layering was dramatically disturbed	Ataxic behavior	[37]
Disorganization of BG population	Knockout mouse with inactivation of the gene coding for the ubiquitin ligase Huwe1 in cerebellar GC precursors and radial glia	GCs migration defects; ectopic GC clusters; layering aberrations	Postnatal lethality	[38]
Radial glia fail to transform into BG	Deletion of *Ptpn11* gene in the entire mouse cerebellum	Disorganized lamination and absence of cerebellar folia	Ataxic behavior	[39]
Deficits in BG specification	Mouse model lacking *Zeb2* in cerebellar radial glia	Compromised inward migration of GCs, cortical lamination dysgenesis	Ataxic behavior	[40]
Ectopic positioning and aberrant stellate phenotype of BG arising postnatally	Conditional knockout mice in which the *APC* gene is inactivated in GFAP-expressing cells	Loss of PCs and cerebellar atrophy	Ataxic behavior	[41]
Decreased numbers, ectopic positioning and aberrant stellate phenotype of BG arising postnatally	Conditional *Sox2* ablation in the whole mouse cerebellum or astrocytes	Cerebellar vermis hypoplasia, abnormal glutamate transport	Ataxic behavior	[42]
BG abnormal development	*Vimentin* knock-out mice	Abnormal PCs dendrites; PCs degeneration	Ataxic behavior	[43]
Immature/reactive-like phenotype acquisition by astrocytes	Cre-loxP conditional deletion of *Dicer* selectively from postnatal astroglia in mice	Cerebellar degeneration, apoptosis of GCs, degeneration of PCs	Ataxic behavior, seizures, uncontrollable movements and premature death arising late postnatally	[44]

BG, Bergmann glia; GC, granule cells; hGFAP, human Glial Fibrillary Acidic Protein, PCs, Purkinje cells.

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
