# Peer review of "Cerebellar Astrocytes: Much More Than Passive Bystanders In Ataxia Pathophysiology"

_jcm, 2020, doi:10.3390/jcm9030757_

Round 1
Reviewer 1 Report
This is a very nice work on astrocytes and ataxias.
My comments are minor:
-introduction:
the cerebellar syndrome is now divided into 3 components: cerebellar motor syndrome, vestibulocerebellar syndrome and cerebellar cognitive affective syndrome (https://www.ncbi.nlm.nih.gov/pubmed/31789706). This is particularly relevant because the anatomy follows this subdivision
-the link between ethanol, retinoic acid and cerebellar astrocytes should be detailed further. Astrocytes are the predominant source of postnatal retinoic acid synthesis in the cerebellum (Brain Res Dev Brain Res 2004;153:233)
-missing: a general scheme of the cerebellar circuitry showing the cerebellar structures involved in the models discussed
-what is the role of astrocytes in the cerebellar reserve?
Author Response
"Please see the attachment."

Reviewer 2 Report
In this review authors discuss astrocytes and neurons in ataxia.
This review provides interesting information, however there are any suggestions on this manuscript.
(1) Author should describe about relationship other glial cells (microglia, oligodendrocytes, NG2glia) and cerebellar atrophy and Purkinje cells degeneration in "Introduction" section.
(2) Until now, it has been reported that there are two types of reactive astrocytes (A1-type and A2-type). Could you add any information about two types of astrocytes in "4.3." section.
(3) Author should correct the word "disfunctions" (Line75, 395, 396).
